

# Resources available for autism research in the big data era: a systematic review

Reem Al-jawahiri and Elizabeth Milne

Department of Psychology, University of Sheffield, Sheffield, United Kingdom

## ABSTRACT

Recently, there has been a move encouraged by many stakeholders towards generating big, open data in many areas of research. One area where big, open data is particularly valuable is in research relating to complex heterogeneous disorders such as Autism Spectrum Disorder (ASD). The inconsistencies of findings and the great heterogeneity of ASD necessitate the use of big and open data to tackle important challenges such as understanding and defining the heterogeneity and potential subtypes of ASD. To this end, a number of initiatives have been established that aim to develop big and/or open data resources for autism research. In order to provide a useful data reference for autism researchers, a systematic search for ASD data resources was conducted using the Scopus database, the Google search engine, and the pages on 'recommended repositories' by key journals, and the findings were translated into a comprehensive list focused on ASD data. The aim of this review is to systematically search for all available ASD data resources providing the following data types: phenotypic, neuroimaging, human brain connectivity matrices, human brain statistical maps, biospecimens, and ASD participant recruitment. A total of 33 resources were found containing different types of data from varying numbers of participants. Description of the data available from each data resource, and links to each resource is provided. Moreover, key implications are addressed and underrepresented areas of data are identified.

## INTRODUCTION

Recently, there has been a move towards generating 'big' and 'open' data in many areas of science such as psychology, neuroscience, genetics, and omics. The move is driven by many stakeholders involved in the academic research process, including funding bodies, publishers, researchers, and participants. Funding bodies (including the Medical Research Council, the Wellcome Trust, and the US National Institutes of Health (NIH), among others) are increasingly funding large studies and initiatives that facilitate data sharing. In addition, their data-policies are recommending and, in certain cases, requiring the studies that they fund to submit a data-sharing plan and to share their data. Publishers are also playing a role in driving big data and data sharing efforts through their data policies, data-oriented special issues and call for papers (e.g., *Focus on Big Data, 2014*; *Eickhoff et al., 2016*), and the launch of machine-readable data journals (e.g., Scientific Data and GigaScience). Depending on the data type and the journal, researchers are expected to

Corresponding author
Reem Al-jawahiri,
rabdalsahib1@sheffield.ac.uk

make their data available either at the time of publication or after an embargo period. Researchers are also playing a role by getting involved in data-sharing initiatives. Some are increasingly preparing their data for sharing (e.g., through proper data management and annotation) and/or requesting access to data from others for secondary analysis. Although participants have concerns about data-sharing policies, many are also supportive of this drive and are participating in studies and initiatives that ask them to consent for their data to be used and shared somewhat flexibly to different researchers and studies (*Oliver et al., 2012*; *Cummings, Zagrodney & Day, 2015*).

'Big data' is an evolving term and its use to refer to data in the mentioned areas is relatively new. Therefore, in this review, the term 'big data' is used to broadly refer to datasets that are larger than what researchers conventionally handle in their respective fields. While 'open data' refers to data that is available for researchers to access without a fee or with a reasonable fee relative to the many costs of collecting data (e.g., funds, time, resources, etc.)

One area where big data is particularly valuable is in research relating to complex heterogeneous disorders such as Autism Spectrum Disorder (ASD). ASD is a heterogeneous neurodevelopmental disorder (or spectrum of disorders, (see *Geschwind & Levitt, 2007*)) that is diagnosed based on a dyad of core behavioural symptoms of social communication deficits and characteristic repetitive and restricted behaviour (*American et al., 2013*). Although these two core symptom domains somewhat unify and define ASD, there is significant variability between people with ASD in terms of symptom severity, symptom profile, level of cognitive function, and comorbidities (*Hewitson, 2013*). This heterogeneity is potentially a major factor impacting on the rate of replication of ASD studies, and is leading some researchers to "give up on a single explanation for autism" (*Happé, Ronald & Plomin, 2006*) and others to propose the possibility that ASD should not be considered as a single disorder (*Geschwind & Levitt, 2007*). Instead, they suggest that within ASD, there could be groups of distinct disorders with many aetiologies (*Geschwind & Levitt, 2007*). Given that the cause, the aetiology and the genetic landscape of ASD remain unknown, developing a clear picture of exactly what ASD is, and what it is not, is increasingly important.

Indeed, ASD is phenotypically heterogeneous between and within affected individuals who show varying trajectories and response to treatment (*Fountain, Winter & Bearman, 2012*; *Gotham, Pickles & Lord, 2012*; *Venker et al., 2014*). Heterogeneity is also implied in ASD's genetic architecture: non-overlapping genetic risk factors are continuing to reveal locus and allelic heterogeneity. Many different genes are involved with ASD and the genetic findings seem to vary depending on numerous factors such as the sample's phenotypic profile, gender (*Werling & Geschwind, 2013*), pedigree (e.g., simplex or multiplex families) (*Sebat et al., 2007*; *Marshall et al., 2008*), and sample size, etc. With the advent of next generation sequencing techniques, the number of genes found that are associated with ASD is increasing to over 800 genes (*Banerjee-Basu & Packer, 2010*); consequently, it is becoming even more challenging to find unified explanations and functional associations between the genes involved. Thus, the unique nature of ASD's heterogeneity and the current state of autism research highlights the importance of

investigating the possibility that ASD may be a disorder with different subtypes, which have distinct phenotypic, cognitive, and/or genetic profiles.

A number of studies have attempted to define the subtype structure of ASD by searching for empirically derived clusters of participants within relatively large datasets. The majority of work in this area has taken a phenotype-first approach, i.e., by attempting to identify potential ASD subtypes by clustering participants based on behavioural, cognitive or medical characteristics (*Cholemkery et al., 2016*; *Georgiades et al., 2013*; *Doshi-Velez, Ge & Kohane, 2014*). In some cases, attempts have been made to demonstrate the convergent validity of these potential subtypes using genetic data (*Liu et al., 2011*; *Veatch et al., 2014*). However, this work has yielded varying findings with regard to the number of identified subgroups, the classification of phenotypes, and the resulting identified genes or loci. For example, the number of proposed potential subgroups varies from two (*Veatch et al., 2014*; *Vieland et al., 2011*), three (*Cholemkery et al., 2016*; *Wiggins et al., 2012*), four (*Doshi-Velez, Ge & Kohane, 2014*; *Liu et al., 2011*; *Veatch et al., 2014*; *Vieland et al., 2011*; *Wiggins et al., 2012*; *Kim et al., 2016*; *Hu & Steinberg, 2009*), to six (*Greaves-Lord et al., 2013*). In addition, there is no agreed description of subtype markers for ASD as there are many relevant possible categories and specifiers (*Lai et al., 2013*); potential candidates include: symptom severity (*Cholemkery et al., 2016*; *Georgiades et al., 2013*; *Veatch et al., 2014*; *Wiggins et al., 2012*; *Hu & Steinberg, 2009*), insistence on sameness scores (*Hus et al., 2007*; *Bishop et al., 2013*), IQ scores (*Vieland et al., 2011*; *Liu, Paterson & Szatmari, 2008*), sensory features (*Ausderau et al., 2014*; *Uljarević et al., 2016*), and ASD comorbidities (*Doshi-Velez, Ge & Kohane, 2014*). Although stratification attempts have been shown to improve statistical power in individual genetic studies, e.g., leading to higher LOD scores in linkage studies (*Liu, Paterson & Szatmari, 2008*; *Buxbaum et al., 2001*; *Shao et al., 2003*), replicable findings are somewhat rare such that different studies find different sets of ASD-implicated genes and loci (e.g., *Liu, Paterson & Szatmari, 2008*; *Buxbaum et al., 2001*; *Shao et al., 2003*). In addition, *Chaste et al. (2015)* found that phenotypic stratification had only a small effect on increasing power and the homogeneity of genetic findings. However in this case phenotypic stratification was based on rather broad, and univariate characteristics such as symptom severity, IQ and diagnostic category, which, in isolation, may not be representative subtype markers.

Contrary to the phenotype-first approach, the genotype-first approach is driven by a reverse strategy in which participants are grouped and studied based on a shared genetic etiology (e.g., a specific CNV or gene associated with a disorder) as opposed to a common phenotypic or clinical profile (e.g., language impairment, ASD diagnosis) (*Stessman, Bernier & Eichler, 2014*; *Stessman, Turner & Eichler, 2016*). Following the identification of a genetically homogenous group, assessments are made to deeply characterize the respective group (i.e., phenotypic characterization and otherwise) and understand its association with the relevant disorders if applicable. Among the recent studies related to the ASD subtyping literature are those supported and facilitated by the Simons Foundation Autism Research Initiative (SFARI), which adopted the genotype-first approach to studying individuals with 16p11.2 CNVs (a recurrent CNV significantly associated with ASD) (see *Simons Foundation, 2016*; *Consortium, 2012*). Deep characterization of 16p11.2 CNV carriers whether through

examining their phenotypic and clinical features (e.g., *Hanson et al., 2015*; *Steinman et al., 2016*), electrophysiological brain activity as recorded with M/EG (e.g., *Jenkins et al., 2016*; *Leblanc & Nelson, 2016*), or otherwise (e.g., *Qureshi et al., 2014*; *Chang et al., 2016*) would ultimately establish whether a particular genetically homogenous group could be regarded as an empirically-derived ASD subtype. As of yet, the ASD subtyping literature, phenotype- and genotype-centric, is growing and advancing in its approach and methodology to investigating ASD potential subtypes; this is largely due to consortia and data resources sharing ASD and ASD-related data to the research community.

Although this is not meant to be a comprehensive review of the ASD heterogeneity/ subtyping literature, from the aforementioned studies and findings it could suggest that successful identification of potential ASD subtypes, if they are to be found, requires data from large numbers of participants who are drawn from a wide sample of the autism spectrum. The task of assembling such a cohort may often be unfeasible for a single researcher, or team of researchers, to accomplish. Therefore, motivated, in part, by the role that big, and open, data can play in defining the typology of ASD, and in understanding the neural aetiology and genetic origin of ASD, a number of initiatives have been established, primarily in the USA, that aim to develop big data resources for autism research. One example is SFARI, which is an ASD data resource that shares data from large numbers of specific phenotypic and genetic profiles of participants, and allows different researchers with different expertise to conduct various analyses on the same sample (e.g., *Hudac et al., 2015*). In addition to the large datasets provided by SFARI, numerous other open ASD datasets exist for use by the research community.

The aim of this article is to identify and describe available resources that provide access to data obtained from participants with ASD. Several existing reviews have accumulated a general list of available data resources, although these are not ASD-specific (*Van Horn et al., 2004*; *Van Horn et al., 2005*; *Van Horn & Toga, 2009*; *Ferguson et al., 2014*; *Poldrack & Gorgolewski, 2014*). Although useful, these lists are non-comprehensive and are mainly limited to neuroimaging resources. In addition, some of the lists included resources that are inaccessible to researchers (e.g., limited access data to only respective consortia members), and thus would not be beneficial to the research community at large. Most importantly, to date, no detailed list of available sources of ASD data exists. Therefore, in order to provide a useful data reference for autism researchers, a systematic search was conducted, and the findings were translated into a comprehensive list focused on ASD data. A total of 33 resources were found containing different types of data from varying numbers of participants. Description of the data available from each data resource, and links to each resource is provided. Moreover, key implications are addressed and underrepresented areas of data are identified.

## METHODOLOGY

Our aim was to conduct the equivalent of a systematic review, which in this paper is described as a 'systematic search'. Therefore, wherever possible, the search for ASD resources closely followed the guidelines and the layout of a systematic literature review, and

**Table 1** Search terms.

| Keywords | Database/Search engine | # of results on primary search |
|---|---|---|
| Autism AND (database OR databank OR repository) | Scopus | 1,050 results |
| Autism AND (database OR databank OR repository) | Google[a] | ~14,400,000 results |
| Autism AND data AND sharing | Scopus | 80 results |
| Autism AND data AND sharing | Google[a] | ~6,310,000 results |
| Autism AND (database OR databank OR repository) AND (EEG OR MEG) | Scopus | 19 results |
| Autism AND (database OR databank OR repository) AND (EEG OR MEG) | Google[a] | ~515,000 results |
| Autism AND (database OR databank OR repository) AND MRI | Scopus | 23 results |
| Autism AND (database OR databank OR repository) AND MRI | Google[a] | ~354,000 results |
| Autism AND (database OR databank OR repository) AND (genetic OR genome) | Scopus | 315 results |
| Autism AND (database OR databank OR repository) AND (genetic OR genome) | Google[a] | ~493,000 results |
| Autism AND (database OR databank OR repository) AND (phenotype OR phenotypic) | Scopus | 145 results |
| Autism AND (database OR databank OR repository) AND (phenotype OR phenotypic) | Google[a] | ~524,000 results |

**Notes.**
[a]Only pages one to three (inclusive) were assessed.

the methodology followed a rigorous structure and search process. The search strategy was performed by the author, Reem Al-jawahiri. The principal focus of the search was datasets containing data pertaining to symptom severity and symptom profile (e.g., scores on the Autism Diagnostic Observational Schedule (ADOS), or Autism Diagnostic Inventory-Revised (ADI-R), cognitive ability, neuroimaging and biospecimens. We used a number of key-word combinations to identify these resources (see Table 1). While carrying out the search we also identified a number of sources of available genetics and omics data; some of which contained data from participants with ASD, others contained genetic or omics data useful for the study of ASD. Where information about these datasets was readily available on the relevant webpage these data sources have also been included and described here. However, where information about the datasets was less clear, either because it was not readily available via the website, or because it required specialist knowledge to understanding the nature of the material held by the dataset, these resources were not included.

## Search terms

The combinations of search terms used, and the number of results returned from either Scopus or Google for each search term are presented in Table 1.

## Inclusion and exclusion criteria

Available data resources that share human ASD data of numerous types including phenotypic, neuroimaging, human brain connectivity matrices, human brain statistical maps, biospecimens, and ASD participant recruitment services were considered eligible for inclusion. Data types, including datasets, databases, databanks, data repositories, institutions offering subject recruitment services, and big-data-initiatives that are currently sharing or have made a public statement of intent to share ASD data at a point in time, were considered eligible regardless of methodology and modality of data collection and analysis. Resources of electronic medical records, clinical trial registries, and data relating to animal studies were excluded. In order to avoid redundant data and clutter, if more than one source housed the same ASD dataset then only the original dataset source or key website was included. Within Scopus, the title, abstract, and keywords of the returned articles were searched using the key search terms. Only articles or webpages written in English were included. The search was carried out between January and March 2016, therefore any new data resources or initiatives made available after March 2016 are not included.

## Search process

The Google search engine and the Scopus electronic database were systematically searched for relevant ASD data resources. The search terms and the different search term combinations used are listed in Table 1. Because Google is a search engine (and not a database with limited results), only pages one to three, inclusive, were assessed; After the third page, the links become less relevant and therefore the limit was set to comprise the first three pages. Naturally, in the case of Scopus searches, all results were assessed. The results in Scopus are based on searches of the search terms in the title/abstract/keywords of the identified journal articles.

Relevant metadata articles, bioinformatics articles, and relevant data portals were also identified and were useful to further complete the data search. Some articles described the data resources or datasets, while others presented tables of useful resources for data access and sharing - although not specifically referring to ASD data (i.e., *Van Horn et al., 2004*; *Van Horn et al., 2005*; *Van Horn & Toga, 2009*; *Ferguson et al., 2014*; *Poldrack & Gorgolewski, 2014*). The Neuroimaging Informatics Tools and Resources Clearinghouse, Neuroscience Information Framework, and SciCrunch are among the data portals and registries used in the systematic search for relevant data resources. Further articles/data resources were found through cited reference searching and through the 'related documents' option in the Scopus database.

The 'information for authors' section of a number of journals, including Nature publishing group journals, Scientific Data journal, PLOS journals, BioMed Central/ GigaScience journal were examined for whether requirements or recommendations were expressed regarding the assimilation of data via supplementary information or via particular data resources/repositories. These journals provided lists of recommended data resources for particular data types for authors wishing to submit their manuscripts and related data. The 'policy' tab of the Wellcome Trust provided a similar, useful list of data resources.

Based on titles, potentially relevant articles or webpages were identified and grouped in tabs, before either the abstract or the webpage of all the saved tabs was assessed. Articles and webpages that consisted of items fitting the inclusion criteria were further assessed after accessing the full text versions and finding the relevant websites of the data resources. Some data providers were contacted for further verification regarding the data type and availability. Duplicates of links leading to the same data resources found from searches, were removed. Similarly, as mentioned earlier, websites hosting the same ASD datasets or overlapping datasets were removed to avoid clutter.

## RESULTS

A total of 33 data resources that provide ASD data were identified. Links to these resources, and a description of the data they hold can be seen in Tables 2 and 3 (in addition to the supporting information in File S1). In both tables, the resources are ordered based on two criteria and the first criterion takes precedent over the second. The first criterion is whether they specialize in only ASD data as opposed to also providing data from other populations in addition to the ASD data. For example, for Table 2, since the National Database for Autism Research (NDAR), SFARI, Autism Genetic Resource Exchange (AGRE), Autism Spectrum Database-UK (ASD-UK), and Interactive Autism Network (IAN) all specifically and solely provide ASD data, they are positioned at the top of the list. The second criterion is based on the similarity (among the resources in the respective table) in the general ASD data-type they handle and provide. For example, Autism Brain Imaging Data Exchange (ABIDE) and the Australian EEG Database (AED) both mainly provide neuroimaging ASD data, therefore these two resources are positioned close to each other.

The 33 identified resources are presented in two separate tables (18 resources in Table 2 and 15 resources in Table 3) based broadly on the category of data type that the resources hold. The resources in Table 2 contain various types of data including phenotypic data, neuroimaging data, biospecimens, brain connectivity matrices, and brain statistical maps. The table also includes data providers that offer ASD participant recruitment services (i.e., ASD-UK and IAN). Four further resources (i.e., Neuroimaging Informatics Tools and Resources Clearinghouse—Image Repository, the LONI Image and Data Archive, COINS Data Exchange, Neuroinformatics Database, Longitudinal Online Research and Imaging System) were identified but were omitted due to them providing overlapping ASD datasets from the same source: ABIDE datasets. Therefore, only the ABIDE website (fcon_1000.projects.nitrc.org/indi/abide) is included in Table 2.

As can be seen in Table 2, data size and participant numbers vary among the resources and data modalities. Moreover, certain resources provide data from large scale studies such as data from SFARI and ABIDE, where all participants take part in the same protocol, whereas other resources such as NDAR and Dryad provide data from very small studies, e.g., studies containing data from 2, 12, 13 and 34 participants with ASD. Additionally, some resources are based on initiatives and collaborations that aimed to collect primary data (not from prior studies) for the purpose of sharing it to the scientific community. For example, AED is a resource with EEG data from over 20,000 patient records

Al-jawahiri and Milne (2017), *PeerJ*, DOI 10.7717/peerj.2880

**Table 2    A comprehensive list of ASD data resources.**

| URL | Resource | Data type category | Data type | Number of participants with ASD |
|---|---|---|---|---|
| ndar.nih.gov | National Database for Autism Research (NDAR)[a] | Phenotypic, neuroimaging, genetic, omics | Phenotypic, neuroimaging, genetic, omics data | Over 80,203 participants (however this number includes the control participants of the ASD studies). |
| sfari.org | Simons Foundation Autism Research Initiative (SFARI)[a] | Phenotypic, neuroimaging, genetic | Phenotypic data, biospecimens, genetic data, neuroimaging data, participant recruitment (to recruit SSC families for additional studies) | Over 3,000 participants (SSC), over 200 participants (Simons VIP), 50,000[b] participants (SPARK). |
| research.agre.org/program/descr.cfm | Autism Genetic Resource Exchange (AGRE)[a] | Phenotypic, genetic, biospecimens | Phenotypic data; genetic data, biospecimens | Over 1,700 families with over 3,300 ASD participants. |
| iancommunity.org/cs/for_researchers | Interactive Autism Network (IAN)[a] | ASD participant recruitment services | Phenotypic data, ASD participant recruitment services | Over 17,000 participants. |
| asd-uk.com | Autism Spectrum Database-UK (ASD-UK)[a] | ASD participant recruitment services | Phenotypic data, ASD participant recruitment services | Over 3,000 families. |
| autismbrainnet.org | Autism BrainNet | BioBank | Postmortem brain and related biospecimens | Over 25 donations (since 2014).[b] |
| fcon_1000.projects.nitrc.org/indi/abide | Autism Brain Imaging Data Exchange (ABIDE) | Neuroimaging | Resting state functional magnetic resonance imaging (R-fMRI), structural MRI, phenotypic data | 539 participants (ABIDE I), 487 participants (ABIDE II). |
| – | Australian EEG Database (AED)[c] | Neuroimaging | EEG data | 50 participants.[d] |
| Brainmap.org | BrainMap[e] | Human brain statistical maps | fMRI, PET, and structural coordinate-based results $(x,y,z)$ in Talairach or MNI space | 70 results/articles relevant to ASD functional data (search using BrainMapWeb). |
| neurovault.org | NeuroVault | Human brain statistical maps | Unthresholded statistical maps, parcellations, and atlases produced by MRI and PET studies | Five studies: 277, 60, 50, 13, 218 participants in each study. |
| umcd.humanconnectomeproject.org | USC Multimodal Connectivity Database | Brain connectivity matrices | Brain connectivity matrices of fMRI and DTI | 42 (fMRI) participants, 51 (DTI) participants. |
| datadryad.org | Dryad | General data repository | lncRNA, MRI, metabolite, MEG | Four studies: two, 34, 12, and 13 participants respectively. |
| figshare.com | FigShare[e] | General data repository | Phenotypic, statistical, genetic data | – |
| nimhgenetics.org | NIMH Repository and Genomics Resource (NIMH-RGR) | Biospecimens, genetic | Biospecimens (DNA samples and cell lines, Induced Pluripotent Stem Cell (iPSC) and Source Cells), GWAS, genomic sequences | Biospecimens: 4,793 families and 19,359 individuals of which 17,189 have DNA cell lines. Genome-Wide Association Studies (GWAS) Data: four studies (1,232 cases, 739 families, 943 families, 935 families). Sequence data (exome): 2,119 cases. |

Al-jawahiri and Milne (2017), *PeerJ*, DOI 10.7717/peerj.2880

**Table 2** (*continued*)

| URL | Resource | Data type category | Data type | Number of participants with ASD |
|---|---|---|---|---|
| bristol.ac.uk/alspac/ | Avon Longitudinal Study of Parents and Children (ALSPAC) | Phenotypic, clinical, biospecimens, genetic | Phenotypic, clinical, biospecimens, genetic (including GWAS, SNPs, VNTRs, in addition to sequence data from UK10K project available via EGA), ALPAC data linked with data (e.g., routine health and social records) from external sources, bespoke data[f] | 96 participants (as identified via follow up questionnaires completed by carers for when the proband was nine years old). |
| catalog.coriell.org | Coriell BioRepositories (including Autism Research Resource) | BioBank | Cell cultures, DNA samples, and induced pluripotent stem cells | 158 ASD cases. |
| neurobiobank.nih.gov | NIH NeuroBioBank (NBB) | BioBank | Postmortem brain and related biospecimens | 64 ASD cases. 22 ASD suspected. |
| kcl.ac.uk/ioppn/depts/bcn/Our-research/Neurodegeneration/brainbank.aspx | Medical Research Council London Neurodegenerative Diseases Brain Bank | BioBank | Postmortem brain and spinal cord tissue | Four ASD cases. |

**Notes.**
[a]Data-type is described in more detail in File S1.
[b]The data is not yet available: It is intended to be available in a future date according to the SFARI website.
[c]There is no website or portal for the AED resource; however, the data is available via email requests to aed@newcastle.edu.au.
[d]The approximate number of ASD participants was found via email correspondence with aed@newcastle.edu.au.
[e]Accurate information regarding the approximate number of participants with ASD is not readily available on the website, due to the nature of the search functionality.
[f]Data specifically from ASD participants are not necessarily available in all the different data types described in this table (therefore further specific enquiries directed to the ALSPAC team is advised).

Al-jawahiri and Milne (2017), *PeerJ*, DOI 10.7717/peerj.2880

Peerj

**Table 3** Genetics and omics data resources either from individuals with ASD or containing data relevant to the study of ASD.[a]

| URL | Resource[a] | Data type category | Data type | Notes |
|---|---|---|---|---|
| mss.ng | MSSNG | Genetic/Genomic | Phenotypic, genomic (whole genome sequencing of blood DNA) | 10,000 participants. However, data from only 3,000 probands is currently available. |
| gene.sfari.org | Simons Foundation Autism Research Initiative Gene (SFARI Gene) | Gene Catalogue | Animal Model, Protein Interaction (PIN), Gene Scoring, CNV | An up-to-date, manually annotated reference set of ASD-linked genes. |
| projects.tcag.ca/autism | Autism Chromosome Rearrangement Database (ACRD) | Gene Catalogue | Genomic structural variation data—CNVs | A curated catalogue of structural variation related to ASD extracted from publicly available literature and unpublished data. |
| autismkb.cbi.pku.edu.cn/index.php | Autism Knowledgebase (AutismKB) | Gene Catalogue | A collection of genes and variations associated with ASD with annotations | – |
| ncbi.nlm.nih.gov/guide/sitemap/ | National Center for Biotechnology Information (NCBI) | Genetics, omics | A collection of multiple resources—omics and sequencing data | – |
| ebi.ac.uk/services/all | European Molecular Biology Laboratory (EMBL-EBI) | Genetics, omics | A collection of multiple resources—omics and sequencing data | – |
| uniprot.org | Universal Protein Resource (UniProt) | Protein sequences | Protein sequences and their annotations | Can be found among EMBL-EBI resources. 91 (reviewed) and 346 (unreviewed) protein records associated with ASD. |
| ebi.ac.uk/ega | The European Genome-phenome Archive (EGA) | Omics—Functional genomics | Interaction of genotype and phenotype (including data from UK10K project) | Can be found among EMBL-EBI resources. |
| thebiogrid.org | Biological General Repository for Interaction Datasets (BioGRID) | Omics | Genetic and protein interaction data | Resource that archives and disseminates genetic and protein interaction data. |
| gpmdb.thegpm.org/index.html | Global Proteome Machine Database (GPM DB) | Omics—Proteomics | Proteomics data from tandem mass spectrometry | Open-source system for analyzing, storing, and validating proteomics information derived from tandem mass spectrometry. |
| peptideatlas.org | PeptideAtlas | Omics—Proteomics | Peptide sequences, mapping—proteome information/data | A collection of peptides identified in a large set of tandem mass spectrometry proteomics experiments. |
| ddbj.nig.ac.jp/index-e.html | DNA DataBank of Japan (DDBJ) | DNA and RNA sequences | DNA and RNA sequences | Annotated collection of all publicly available nucleotide sequences and their translated amino acid sequences. |

Al-jawahiri and Milne (2017), *PeerJ*, DOI 10.7717/peerj.2880

**Table 3** (*continued*)

| URL | Resource[a] | Data type category | Data type | Notes |
|---|---|---|---|---|
| chr7.org | The Chromosome 7 Annotation Project | DNA sequences | DNA sequence and annotation of the entire human chromosome 7 | 84 cases. |
| mirbase.org | miRBase: the microRNA database | miRNA sequences | miRNA sequences and annotation | – |
| gbrowse.csbio.unc.edu/cgi-bin/gb2/gbrowse/slep | Sullivan Lab Evidence Project (SLEP) | Genetics, omics | A collection of genes and variations associated with ASD with annotations | Findings from genome wide linkage (GWL), genome wide association (GWA), and microarray (MA) studies for ASD. |

**Notes.**

[a]The resources listed in this table contain data either from individuals with ASD or data relevant to ASD research that is collected from non-affected individuals (e.g., from individuals with certain genetic profiles or syndromes related to ASD research).

(however only 50 ASD EEG records) made available through the collaboration between John Hunter Hospital and the University of Newcastle, Australia.

The data types of the resources in Table 3 are somewhat more homogenous and would fall under the categories of genomic, genetic, and omics/proteomics data. For example, data types include gene catalogue (e.g., CNVs and protein interactions), sequencing data (e.g., DNA sequencing, RNA sequencing, and protein sequencing), functional genomics data (interaction of genotype and phenotype), genetic and protein interaction data, in addition to other genetic and omics data. The relevant data from these resources are either genetic or omics data shown to be potentially involved with ASD (collected from unaffected participants) or data from participants with ASD. Several resources, namely, NDAR, SFARI, AGRE, and NIMH Repository and Genomics Resource, also provide various genetic data, however these resources are grouped with the resources in Table 2 because they also provide additional data-types such as phenotypic, neuroimaging, and/or biospecimens.

Some of the resources in Tables 2 and 3 such as NDAR, SFARI, AGRE, and MSSNG, are specialized in ASD data and aim to share only autism-related data. While other resources, due to the purpose of the resource or the nature of the data, aim to share data involved with numerous populations and conditions. Nevertheless, these resources also housed data involved with ASD research. Examples of non-ASD specialized resources include Coriell BioRepositories and Universal Protein Resources. The resources vary in the way the data is described or made viewable, searchable, and accessible for researchers. For example, different informatics portals or systems (see the 'URL' columns in Tables 2 and 3, and/or the metadata articles in File S1) are used to achieve the task of data sharing.

NDAR, SFARI, AGRE, MSSNG, IAN, and ASD-UK are among the key resources which provide ASD data. NDAR (*Hall et al., 2012*) is the largest ASD resource housing ASD data of various types from numerous sources. It is a US NIH-funded initiative that aims to "accelerate progress in ASD research through data sharing, data harmonization, and the reporting of research results" (ndar.nih.gov). NDAR shares various types of data: phenotypic data (e.g., clinical and diagnostic assessments such as the Autism Diagnostic Inventory (ADI) and ADOS); neuroimaging data (e.g., MEG, fMRI, and DTI); genetic and omics data (e.g., DNA (Microarray and Sequencing) and RNA (Microarray and Sequencing)) related to human participants. NDAR is evolving and it is now incorporated under the NIMH Data Archive (data-archive.nimh.nih.gov). Researchers can access the data without any fees, however they need to first request and gain approval for access (see *Payakachat, Tilford & Ungar, 2016*, for a flow diagram for data access). The NDAR system enables data to be aggregated not only from other major data resources (e.g., SFARI, AGRE, and IAN), but also from qualified autism researchers regardless of funding source. Currently, NDAR contains data from over 80,000 participants (however this number includes neurotypical control participants of the ASD studies).

SFARI is an ASD data resource that shares very well-characterized participant data due to the rigorous standardization and coordination among the SFARI-funded investigators (currently over 250 investigators). SFARI's aim is to "improve the understanding, diagnosis and treatment of autism spectrum disorders by funding innovative research of the highest quality and relevance" (sfari.org/about-sfari). SFARI also aims "to facilitate and drive

research in the field as a whole" through granting data access to autism researchers (regardless of their funding source). Approved researchers are granted access to the data without a fee, however biospecimen data is available on a modest fee-for use basis. SFARI's data are segmented and collected from three large cohorts: Simons Simplex Collection (SSC), Simons Variation in Individuals Project (Simons VIP), and Simons Foundation Powering Autism Research for Knowledge (SPARK). The SSC is intended to be especially useful for investigating de novo (new) mutations due to the cohort comprising of nearly 3,000 simplex families. Simons VIP contains data from over 200 participants with recurrent genetic variants involved with ASD (i.e., deletion and duplication of chromosomal region 16p11.2 and 1q21.1). SPARK is a new initiative involving collecting phenotypic and biospecimen data from 50,000 participants with ASD. This initiative will later provide researchers with participant recruitment services through retaining a database of the SPARK participants who consent to be re-contacted for future studies.

AGRE is a resource that is mainly supported by Autism Speaks and is among the leading resources committed to advancing ASD genetic research. AGRE shares phenotypic (e.g., ADI-R and ADOS), genetic (e.g., high-density SNP and Genome Wide Association Study data), and biospecimen data (e.g., cell lines, DNA and plasma) of over 1,700 well defined multiplex and simplex families with over 3,300 participants with ASD. AGRE can also provide participant recruitment services. In order to access data, researchers need to request data and pay per sample fee for obtaining biospecimen data.

MSSNG is the largest ASD genomic resource made available to the research community. It is the outcome of the diverse collaboration efforts between Autism Speaks, SickKids Hospital, University of Toronto, and Google. MSSNG's aim is "to provide the best resources to enable the identification of many subtypes of autism, which may lead to better diagnostics, as well as personalized and more accurate treatments" (research.mss.ng). The target is to sequence and share data from whole genome sequencing of blood DNA of 10,000 families affected by ASD. The cohorts consist of AGRE participants, but could later include other well-phenotyped cohorts. Over 5,000 whole genome sequences (and coded phenotype data) with nearly 3,000 sequences from participants with ASD are freely available to approved researchers.

IAN and ASD-UK are both useful resources based in the US and the UK respectively, that enable the collaboration between researchers and participants. With the aim of facilitating and advancing ASD research, these resources provide approved researchers with participant recruitment services and phenotypic data. The data that can be accessed consist of certain phenotypic data (e.g., Social Communication Questionnaire) along with medical data (e.g., medical history). Researchers can request and access these resources at a cost. To date, IAN and ASD-UK registered over 17,000 participants with ASD and over 3,000 families, respectively. For more information on NDAR, SFARI, AGRE, ASD-UK, and IAN in addition to a list of relevant metadata articles, see File S1.

## DISCUSSION

The systematic search showed that there are at least 33 data resources and initiatives that aim to share available ASD data to interested researchers, and therefore further advance ASD

research and understanding of its heterogeneity and potential subtypes. These resources and initiatives have undoubtedly had, and will continue to exert, significant influence on the field of ASD research. Particularly, notable initiatives such as NDAR and SFARI can, arguably, have an impact on the nature of ASD research and the process of data sharing. Although NDAR and SFARI share similar aims, they differ in their approach and thus affect the field differently. Both resources aim for the advancement of ASD research through supporting certain ASD projects and facilitating the secondary analysis of data. Additionally, NDAR and SFARI are specialized ASD resources that aim to accommodate for ASD-specific data types and methodologies. As a result, these resources led to new findings and numerous publications in the field. Each resource, however, accomplishes its aims through a distinctive approach and data sharing process, which arguably differentially influences the nature of ASD research and data sharing process. Specifically, each resource adopted a different approach to obtaining its primary data. Even though NDAR and SFARI initially obtained and shared data collected solely from particular projects that they funded, NDAR currently obtains its primary data from multiple sources including over 120 NIH-funded investigators in addition to other autism researchers irrespective of funding source. SFARI, on the other hand, obtains its primary data only from SFARI-funded investigators mainly involved in large projects. One approach (i.e., NDAR's approach) interacts with various sources that range in the scale of their data, while the other interacts with sources that offer large datasets. Consequently, the former approach supports the secondary analysis and sharing of smaller scale data, while the latter drives the reuse of large datasets and the sharing of large data.

Both approaches have their merits and there are advantages and disadvantages of the sharing of either small or big data. One advantage of sharing small data is to allow the scientific community to make use of the rich variety of research data that is collected by small independent research efforts. As ASD is a clinical population that is typically challenging to recruit and examine, especially in its more severe form, most ASD studies therefore are comprised of small sample sizes or small data. Collectively, these small efforts would make up a large sum of studies, with data from various specialties and modalities. This can be valuable for secondary analysis if the data is shared, especially if the datasets can either be analysed individually or integrated into different combinations. Whether to augment the scale of the dataset or to explore links between data from different modalities and levels, integrating datasets from different studies is a potentially greatly beneficial yet challenging task that some data resources are tackling. A model initiative and collaboration between NDAR and SFARI towards this direction resulted in the development of a tool that ultimately helps in identifying overlapping data and in linking data from different studies. The tool is a free software application that assigns global unique identifiers (GUID) to research participants (see *Johnson et al., 2010*, GUID Tool (https://ndar.nih.gov/tools_guid_tool.html), *Rudacille, 2010*). Using an algorithm derived from four input variables taken from a participant's birth certificate, the GUID tool produces a sequence of letters and numbers to serve as a standardized, unique, and anonymized code specific to a research participant, regardless of the study the participant is engaged in. The generated GUID will always be the same for the particular participant and
thus will be used across studies if the participant participates in more than one study. This system would make it possible to trace data across studies of the same participant and would facilitate data overlap detection, especially if more studies and data resources use GUIDs. Hence, NDAR requires all prospective studies to include GUIDs in the data submission plan (and encourages retrospective studies to include GUIDs or pseudo-GUIDs). Other than NDAR and SFARI, a number of major ASD resources, namely, IAN, AGRE, and Autism BrainNet generate GUIDs for their studies making GUIDs an even more effective tool.

However, despite these efforts, combining or linking small different datasets remains a difficult task requiring sophisticated integration tools and the participation and collaboration of data resources. While in the case of big data, the datasets offered are large homogenous datasets that although could also benefit from being combined with other datasets, it is in itself ready for re-analysis with great statistical power. Despite the challenge of integrating small data, sharing data (in both cases small or big) would encourage more transparency and rigorous research, whether because the re-analysis would uncover limitations or errors not previously identified; or because the prospect of sharing data, encourages the respective researcher to be more cautious in their analysis and/or reporting. This can be implied from the findings of a recent report, which found that psychology studies that were unwilling to share data were associated with weaker statistical evidence, and more errors in statistical reporting of results in comparison to studies with shared data (*Wicherts, Bakker & Molenaar, 2011*).

Other than the impact of the available ASD resources, a second point to consider is the role of retrospective data and whether it has a role to play towards the shift to big data and open data. Retrospective data here is referred to data that is previously collected by researchers, clinicians, or other entities for the purpose of the respective studies or for keeping medical records and/or other information about patients or groups of interest. Naturally, there is a large sum of ASD retrospective data whether cumulatively from small independent studies or from clinical sources with many records and information. Although invaluable for the advancement of ASD research and for medical purposes, this type of data is collected without taking into account the prospect of sharing it. If shared, this ASD data can be used by researchers for secondary or primary analysis, respectively. However, the sharing of data that is not initially prepared for the purpose of sharing can be challenging and the available data-sharing resources lack initiatives that address this challenge. The resources should tailor to retrospective data and aim to integrate and present annotated ASD data that later gained permissions for sharing. NDAR along with some EEG-specialized and general-purpose resources, such as AED, Dryad, and FigShare, among others, allow for the sharing of certain retrospective data, however tailored resources that cater to retrospective data from various sources and modalities is still limited.

A third point to consider is the difficulty of finding certain ASD data and sufficient data annotation that are of interest to researchers. As shown in Tables 2 and 3, and the File S1, data from certain modalities are underrepresented or even lacking. For example, none of the available resources provide access to eye-tracking data obtained from participants with ASD. In addition, very few resources provide EEG or MEG data obtained from participants with ASD: despite the large number of datasets being held by NDAR, no EEG datasets

and only 21 MEG datasets are available from individuals with ASD. Even when this type of data (or neuroimaging data generally) is found, researchers face other difficulties in finding certain information about the data. One bit of data information that is essential to researchers is pertaining to the level of data quality (both the raw and processed data) in accordance with the standards of the specific field and modality type. However, a system that describes quality is not developed and the data-sharing process, typically, does not permit researchers to have a quick view of the actual data. If data can be viewed (before it is officially obtained), researchers would be better able to determine the quality and feasibility of the data in relation to the researchers' interests, skills, and resources. The available ASD data websites either describe or display the available data (or both), however it can be challenging at times to find specific information relating to sample size and other details about the data. For example, it is unclear in certain cases whether the mentioned sample size, relating to an available dataset, is that of solely participants with ASD or whether it is inclusive of control participants. Furthermore, information about the data that will be shared from on-going studies is very limited if at all reported. Generally, although the current data-websites describe or annotate the data in a certain way, further comprehensive and clear annotation is necessary along with a better inclusion of data from different modalities.

## CONCLUDING REMARKS

Here, we have identified 33 resources that provide data obtained from individuals with ASD to the research community. While we are confident that we have identified the vast majority, if not all, of the publically available data sources that provide behavioural and neuroimaging data from individuals with ASD, we may not have captured some resources in the cases where research groups have data that they are willing to share, but did not publicize the data in a way that was identifiable via our key-word search. The resources and data-websites are not static: they are improving and the number of available data is increasing in size and variety. Notably, the drive for big data, sharing data, and for accelerating research on complex heterogeneous disorders such as autism is not only leading to the on-going growth of current initiatives, but also of the development of new ones, e.g., MSSNG.

### Funding

The authors received no funding for this work.

### Competing Interests

The authors declare there are no competing interests.

### Author Contributions

- Reem Al-jawahiri conceived and designed the experiments, performed the experiments, analyzed the data, wrote the paper, prepared figures and/or tables.
- Elizabeth Milne conceived and designed the experiments, reviewed drafts of the paper.
## Data Availability

The sources of the raw data evaluated are listed in the article.

## Supplemental Information

Supplemental information for this article can be found online at http://dx.doi.org/10.7717/peerj.2880#supplemental-information.

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
