# Peer review of "Resources available for autism research in the big data era: a systematic review"

_PeerJ, doi:10.7717/peerj.2880_

## Round 0.1 · original submission · Minor Revisions

Thanks for submitting this interesting and potentially very useful paper to PeerJ. Like Reviewer 2 I was a little unsure whether it was within the scope of the journal but after discussion with the journal editors, I’m happy to invite a resubmission pending Minor Revisions.

Both reviewers suggest additional databases that could be included. In each case, you should either add the database to the revised manuscript or provide a rationale for its omission in your response.

Reviewer 1 suggests that you set up a website so that the information in this paper can be kept up-to-date. I think this is a great idea but it may take a while and shouldn’t hold up publication of this paper (I don’t think Reviewer 1 is suggesting that it should).

Reviewer 2 requests a citation for the claim that participants support data sharing. Just after reading the review, I stumbled across the following paper, which might be relevant http://journals.plos.org/plosone/article?id=10.1371/journal.pone.0125208

Personally, I think the most useful part of the paper might be Supplementary File S1, which provides information about all the measures in the “big” databases in which the same measures have been used consistently across studies. It’s a shame these tables are too big to go in the paper itself, but it would certainly be helpful if you could consider the following small amendments:

1. Incorporate the information about sample size in File S1 so that readers don’t have to flick back and forth between the paper and the supplement.

2. Use consistent names for the same measures across the different databases (e.g., you currently use ADI, ADI-R, Autism Diagnostic Interview)

3. Indicate whether item-level data are available or just subscales or total scores. For example, phenotypic subtyping has been attempted based on ADI-R data, but this is only possible if responses to individual items on the ADI-R are available.

One final issue that might be worth discussing in the paper is the GUID unique identifier system that some databases are or at least were using in order to link together data from different sources for the same participants and weed out repetitions https://ndar.nih.gov/tools_guid_tool.html
https://spectrumnews.org/news/researchers-debut-unique-identifiers-for-study-participants/

·

Basic reporting

The introduction does an overall good job of introducing the topic of heterogeneity in autism and the obstacles it presents to the field. However, there are some things that may help to further improve this section. For example, the Lai et al., 2013 paper in PLoS Biology on subgrouping the autism spectrum and discussion of this topic in the context of what was then the recent changes from DSM-IV to DSM-5, seems highly relevant to this topic, but is currently not cited in the manuscript. Grzadzinski, Huerta, & Lord (2013) also discuss DSM-5 and opportunities for subtyping that is not cited. There are other studies on developmental trajectories and differing outcomes/prognosis that aren’t cited as well (e.g., Pickles, Anderson, & Lord, 2014, JCPP; Lord, Bishop, & Anderson, 2015, Am J Hum Genet; Anderson, Liang, & Lord, 2013, JCPP; Anderson et al., 2007, J Consul Clin Psychol; Szatmari et al., 2015, JAMA Psychiatry; Fein et al., 2013, JCPP; Lombardo et al., 2015, Neuron). There are also other prominent studies using approaches like clustering to find subgroups/subtypes, that the authors don’t cite (e.g., Hu & Steinberg, 2009, Autism Res; Hu et al., 2009, Autism Res; Byrge et al., 2015, J Neurosci; Lombardo et al., in press Scientific Reports). I completely understand that the authors are not trying to give a comprehensive overview of all the important studies on parsing heterogeneity, but it is not clear why the authors have been selective in choosing some studies to cite, but not others. Clearly there are a massive number of ways one could try to subgroup the autism spectrum based on known a priori knowledge (a point that the Lai et al., 2013 paper tries to make) and there are also other approaches that are more data-driven, so its not likely that all the relevant work can be cited here. However, perhaps this is a point that needs to be explicitly stated, since if its not stated, it may seem to an uninformed reader that the authors are citing most of the main primary important studies in the literature.

There is also one sentence (lines 62-65) that needs changing. The authors write:

“This heterogeneity, coupled with a low rate of replication of ASD studies, is leading some researchers to “give up on a single explanation for autism” [6] and others to propose the possibility that ASD should not be considered as a single disorder [3].”

This sentence, particularly the usage of the phrase ‘coupled’, implies that heterogeneity is one thing and that lack of replication is another independent thing, and the two are coupled together in autism. This is one view but certainly not the only view. One could posit that the lack of replication is in part because of the prominent heterogeneity. In this case, is ‘coupling’ the right phrase to describe such a relationship? If lack of replication is composed of many factors, and heterogeneity is one big factor explaining why studies don’t replicate, then is the phrase ‘coupling’ the right way to characterize this phenomenon? Perhaps there is a way of arguing that ‘coupling’ is a phrase that is warranted, but upon multiple readings of that sentence, it seems to imply to me that we have two separate phenomena here, and they happen to be correlated somehow. To make an analogy here, if you had a car and the engine of that car, would you say ‘the car, coupled with the engine’, or is it better to conceptually think of it as the engine is a big part of why the car is a car, and simply talk about the car and how a big component of the car that is of importance is the engine.

It also seems pertinent that the authors should move discussion about heterogeneity at the phenotypic and genetic levels into the same paragraph, and before they make statements about the impact of heterogeneity on the field. For instance, in the statement about ‘giving up on a single explanation of autism’ or that there are multiple ‘autisms’ the researchers are citing papers (Happe et al.,; Geschwind & Levitt) that make such statements not just from evidence that autism is phenotypically very heterogeneous, but also the wealth of studies that even at that point in time (2006-2007) already suggested a wide array of possible etiological mechanisms and routes towards developing autism. In my mind, it makes sense to organizationally reframe talk about heterogeneity, by placing statements about phenotypic and genetic heterogeneity within the same paragraph, as its important readers understand that statements about heterogeneity apply at multiple levels of analysis, not just the phenotype.

The 5th paragraph of the introduction (starting at line 83) is somewhat biased towards weighting studies at the phenotypic levels as the majority of work that tries to subtype autism. In my viewpoint, there is a heavy amount of genetics research that parses heterogeneity via a ‘genetics first’ approach, but the authors do not really talk much about this viewpoint in contrast to studies that look at the ‘phenotype first’. It is probably worth noting that different approaches at different levels of analysis may tell us very different things, and its not clear that one is more right than another. There are arguments that could be made about the critical importance of phenotypic subgroups particularly for impact in clinical applications (e.g., making prognosis predictions), while other strong arguments can also be made that genetics first approach yields insight that cannot be seen at other levels of analysis (i.e. finding novel drug targets).

In the same 5th paragraph of the introduction, the paragraph reads as being critical of previous studies and their attempts to subgroup/subtype, claiming inconsistency of number of subgroups and/or lack of agreement on which features are relevant for subtyping. I think being overly critical about these things at such an early stage of the work on this topic is perhaps not helpful, since these kinds of criticisms imply/assume that there is a static correct answer out there and that if research was done correctly that we’d all stumble upon the same finding. Such an assumption doesn’t seem appropriate though (at least at this stage), since subgrouping/subtyping is a research approach that may lead us down to making specific types of stratifications that may or may not be the same across all contexts and circumstances. If the main purpose of subtyping was to make more personalized inferences and insights about specific kinds of patients with autism, and if those subtype distinctions are clinically relevant or help us hone in more sensitively on biological mechanisms, does it necessarily matter if they do not agree from study to study? There must be certain purposes for subtyping that make clinical or translational impact, but nonetheless may differ in how they split up the autism spectrum.

Experimental design

The authors do a good job of making explicit the systematic search that they embarked upon to find these resources, and this thorough description helps enable reproducibility. Since the results of their systematic search do not necessitate further evaluation in the way a typical empirical paper would be evaluated, there is not much more to say with regards to methodology here.

Validity of the findings

The finding the authors report of 32 different data sources is very helpful and potentially quite useful for the autism research community. My only suggestion here is that the authors might find that the best way to have impact on the research community (in addition to communicating their results in a paper like this) is to make a website that could be continually updated by themselves and/or the community, in order to easily direct researchers to data sources such as these, as well as compile other kinds of helpful information about each data source. Such a resource would be helpful, since as time goes on the landscape will ultimately change. Thus, rather than doing a bunch of complex searches like the authors have done here, in the future one should simply have to do a Google search of “autism AND big data resources”, and find that the first hit is this website showing all the known resources.

There is also one suggestion for a data source that the authors do not discuss, and perhaps this is because it did not show up in their systematic search. Gene Expression Omnibus (GEO; https://www.ncbi.nlm.nih.gov/geo/) is a heavily utilized repository for functional genomics research (e.g., transcriptomics), and there are many autism and/or autism-relevant datasets one can find here. As of Oct 7th, if I do a search for “autism” in GEO, I get 3691 hits for GEO Datasets. Perhaps not all of those are actually relevant, but many are. Many labs/teams doing functional genomics work routinely deposit their data on GEO, and thus, in addition to finding many autism gene expression datasets, one could also identify many gene expression datasets from animal models of high relevance for ASD within GEO as well. Since this did not appear in the systematic search, yet is now being suggested by a research within the community, this makes it all the more clear that a useful tool for the community could be a crowd-sourced website/database of autism-relevant data resources like this, in order to aid the community in their work on autism.

Below are a couple other resources that may not necessarily contain data on autism patients per se, but are nonetheless central resources that could be relevant to many different types of investigations in autism research. For example NKI-RS has SRS data in conjunction with lots of imaging data and other deep phenotyping relevant for other disorders all on the same individuals.

NIH Pediatric MRI Data Repository - http://pediatricmri.nih.gov/nihpd/info/index.html
The enhanced Nathan Kline Institute-Rockland Sample (NKI-RS) - http://fcon_1000.projects.nitrc.org/indi/enhanced/
Allen Institute Online Public Resources - https://www.alleninstitute.org/what-we-do/brain-science/research/online-public-resources/
Genotype-Tissue Expression (GTEx) project - http://www.gtexportal.org/home/
Exome Aggregation Consortium (ExAC) - http://exac.broadinstitute.org
Human Connectome Project - http://www.humanconnectome.org
UK Biobank - http://www.ukbiobank.ac.uk

Additional comments

Overall, this paper is an important resource for individuals within the autism research community who may want to capitalize on existing datasets but may not know where to look for such resources.

·

Basic reporting

The paper meets basic reporting standards.

Experimental design

This is a compilation of openly available datasets on autism that could be useful for the research community. The main issue for me is whether it meets the criteria for papers in PeerJ. I am aware that PeerJ is explicit in its Aims and Scope information that it does not publish reviews (something many researchers would like to see changed!); this is not a review of the literature, but it is a review of resources, and as such does not have a well-defined research question.
If the editors agree that the paper must be rejected on these grounds, then there are options that the authors could consider: either a specialist autism journal or possibly a submission to the Meta-Research section of PLOS Biology (though I’m not 100% sure it meets their criteria): http://journals.plos.org/plosbiology/article?id=10.1371/journal.pbio.1002334

Validity of the findings

I had a couple of thoughts about the information in Table 2. First, do we know if there is any overlap in cases across datasets? That would be important when combining datasets or using one to replicate another. Second, for most resources, the number of ASD cases is given, but for Brainmap.org and neurovault.org we are told of the number of studies. It also seemed odd to include Figshare without giving any indication of how many useful datasets were deposited there.
I also wondered why ALSPAC was not included: was this because its data access arrangements are not open enough to meet criteria, or was it not picked up by the search algorithm? If the latter explanation applies, then one wonders whether there might be other whole-population cohorts that had been omitted which could potentially be of value. Although ALSPAC is a whole population cohort, there are cases with autism included, and some papers have been published on them. It is exceptional in terms of the amount of phenotypic data plus DNA.

Additional comments

The paper was generally well-written though perhaps a little repetitive: I felt that by removing redundant material there might be scope for shortening by 200-250 words.
Other than that, I offer just a few minor stylistic comments:
46-48: The claim that participants are supportive of data-sharing needs some evidence: it seems reasonable that many participants are, but others may be concerned about misuse.
52: larger than those
98 merely-> only
104: delete ‘obtaining’
113: delete ‘However’

---

## Round 0.2 · accepted · Accept

Thank you for such a thorough rebuttal. It made the decision very straightforward. Congratulations on what I think will be a very useful paper for researchers in the field.

·

Basic reporting

The revised manuscript addresses all my prior comments in this area.

Experimental design

The revised manuscript addresses all my prior comments in this area.

Validity of the findings

The revised manuscript addresses all my prior comments in this area.

Additional comments

The revised manuscript addresses all my prior comments, and I can recommend accept.